# Systemic Sclerosis-Associated Pulmonary Hypertension: Spectrum and Impact

**DOI:** 10.3390/diagnostics11050911

**Published:** 2021-05-20

**Authors:** Mario Naranjo, Paul M. Hassoun

**Affiliations:** Division of Pulmonary and Critical Care Medicine, Department of Medicine, Johns Hopkins University, Baltimore, MD 21287, USA; mnaranj7@jhmi.edu

**Keywords:** systemic sclerosis, pulmonary hypertension, pulmonary arterial hypertension, pathogenesis, diagnosis, management, prognosis

## Abstract

Systemic sclerosis-associated pulmonary arterial hypertension (SSc-PAH) is a catastrophic complication of one of the most common and devastating autoimmune diseases. Once diagnosed, it becomes the leading cause of mortality among this patient population. Screening modalities and risk assessments have been designed and validated by various organizations and societies in order to identify patients early in their disease course and promptly refer them to expert centers for a hemodynamic assessment and formal diagnosis. Moreover, several large multicenter clinical trials have now included patients with SSc-PAH to assess their response to therapy. Despite an improved understanding of the condition and significant advances in supportive and targeted therapy, outcomes have remained far from optimal. Therefore, rigorous phenotyping and search for novel therapies are desperately needed for this devastating condition.

## 1. Introduction

Systemic sclerosis (SSc) is a chronic multisystem disease characterized by fibrosis, excessive collagen deposition within the skin and internal organs, chronic inflammation, autoimmune dysregulation, and microvascular endothelial dysfunction that ultimately leads to multiorgan dysfunction [1,2]. SSc is traditionally classified based on the extent of skin involvement, the accompanying pattern of internal organ disease, and the presence of overlapping features with other systemic autoimmune diseases. Limited cutaneous SSc (lcSSc) involving the distal skin of the extremities and the face; and diffuse SSc (dcSSc) involving large areas of skin in the proximal aspect of the extremities and multiple organs [3]. Both forms, however, are systemic diseases associated with significant morbidity and mortality. Pulmonary arterial hypertension (PAH, Group 1 of the World Symposium on Pulmonary Hypertension classification) [4] is more common in lcSSc but can complicate other variants of the disease. Moreover, forms of pulmonary hypertension (PH) other than PAH can afflict SSc patients, including PH related to left heart disease, interstitial lung disease (ILD), chronic thromboembolism, and pulmonary venous occlusive disease, which further complicates diagnosis and management [4]. Although many forms of PH can complicate SSc, this review will focus essentially on SSc-associated pulmonary arterial hypertension (SSc-PAH) for which treatment is available. SSc-PAH has emerged as a leading cause of morbidity and mortality. Patients with SSc-PAH have a higher risk of death than patients with idiopathic pulmonary arterial hypertension (IPAH), despite similar hemodynamic indices, and are frequently less responsive to PAH therapy [5,6]. Early diagnosis of PAH and initiation of treatment continue to be challenging in SSc due to several factors ranging from limitations of the current screening strategies and the complexities of the disease.

## 2. Epidemiology

The diagnosis of PAH in SSc, and other associated etiologies, is confirmed when the mean pulmonary artery pressure (mPAP) is greater than 20 mmHg at rest, measured by right heart catheterization (RHC), with a pulmonary capillary wedge pressure (PCWP) less than or equal to 15 mmHg, and a pulmonary vascular resistance (PVR) more than or equal to 3 Wood units [4]. The prevalence of PAH among patients with SSc has been estimated to be between 5% and 15% [7,8,9]. Several studies, including data from national and international registries, have described the incidence and survival rates related to SSc-associated PH and PAH. PHAROS (Pulmonary Hypertension Assessment and Recognition of Outcomes in Scleroderma) [10] represents one of the largest cohorts of SSc patients. This study identifies important risk factors associated with the development of PH and PAH, including diffusing capacity of carbon monoxide (DLCO) <55% predicted, forced vital capacity (FVC)/DLCO ratio >1.6, and/or right ventricular systolic pressure (RVSP) obtained by echocardiography >40 mmHg. Participants included in this study underwent RHC if clinically indicated, revealing that PH had developed in 10% at 2 years of enrollment, 13% at 3 years, and 25% at 5 years; while PAH occurred in 7% at 2 years, 9% at 3 years, and 17% at 5 years [10]. Additionally, REVEAL (Registry to Evaluate Early and Long-term PAH Disease Management), the largest registry of PAH patients in the US, reported that collagen vascular disease (predominantly SSc) was the second most common cause of PAH after IPAH which is more than twice as common as SSc-PAH [11]. Overall, SSc-PAH is the second most frequent cause of PAH in both US and European registries following IPAH [12]. PHAROS also addressed gender and demographics disparities. SSc-PAH occurred more commonly in women (87%), white (67%), and patients who had lcSSc (57%) [10]. With regard to survival, Pasarikovski et al. reported that males with SSc have a shorter mean time to PAH diagnosis (1.7 ± 14 vs. 5.5 ± 14.2 years) and shorter PAH duration (3.5 ± 3.1 vs. 4.7 ± 4.2 years) with worse 5-year survival compared with females (46 vs. 57%, *p* = 0.07) [13]. Patients with SSc-PAH are less responsive to therapy than patients with IPAH [6,14]. Hence, survival is lower in SSc-PAH patients (60% vs. 77%, *p* < 0.001) despite similar baseline characteristics and lower mPAP, PVR, and pulmonary artery systolic pressure (PASP) compared with IPAH patients [5]. These findings emphasize the importance of early SSc-PAH screening, diagnosis, and treatment to avoid complications and improve outcomes.

## 3. Pathogenesis and Clinical Features

There are five groups of PH based on the mechanisms of disease, clinical presentation, hemodynamic characteristics, and therapeutic response. Various forms of PH can occur in SSc patients [4]. PH related to remodeling of precapillary arterioles is classified as group 1 (SSc-PAH), while PH caused by left heart disease (such as valvular disease or systolic or diastolic myocardial dysfunction) is group 2. PH due to chronic hypoxia from advanced ILD is group 3 [15]. The exact mechanisms involved in the development or progression of SSc-PAH remain unclear, but several inflammatory and endothelial pathways are implicated as potential culprits. Moreover, a disequilibrium between vasoactive, proliferative mediators (e.g., thromboxane A2 and endothelin-1), and antiproliferative vasodilators (e.g., nitric oxide and prostacyclin) from endothelial injury and dysfunction, and potentially intraluminal microthrombosis, lead to progressive pulmonary arterial remodeling and increase in PVR [16,17,18]. Increased sympathetic activity, tissue hypoxemia, and ischemia-reperfusion injury to the pulmonary vasculature promotes additional cytokine release, furthering vascular remodeling, fibrosis, and thrombosis [19]. The progressive increase in PVR, pulmonary arterial pressure, and right ventricular (RV) pressure overload eventually overcome the compensatory hemodynamic mechanisms leading to RV failure and death [20,21,22,23,24].

### 3.1. Autoimmune Dysfunction

Autoantibodies more frequently associated with an increased risk of SSc-PAH include anticentromere antibodies (ACA), anti-U1-ribonucleoprotein antibodies (RNP), nucleolar pattern of anti-nuclear antibody (nucleolar-ANA), antiphospholipid antibodies, and the absence of anti-Scl 70 [25,26,27]. In contrast, SSc patients with Scl 70 autoantibodies are more likely to develop PH due to ILD. The presence of anti-RNA polymerase III autoantibodies has been associated with more skin and renal involvement rather than PH [26]. Furthermore, some specific autoantibodies found in SSc-PAH patients are thought to have potential causal implications. Angiotensin II type-1 receptor antibodies and endothelin-1 receptor type A antibodies are upregulated and could function as inflammatory mediators, increasing cytotoxicity and contributing to vascular remodeling [28]. Antifibroblast antibodies, detected in up to 30% of patients with SSc-PAH, trigger several mechanisms vital to the vascular remodeling process, including the activation of platelet-derived growth factor receptors, which stimulates the release of reactive oxygen species, fibroblast proliferation, and collagen synthesis [19,29].

### 3.2. Interstitial Lung Disease-Associated PH in SSc

ILD is common in SSc, with evidence of interstitial changes on imaging in up to 90% and chronic respiratory failure in approximately 10% of patients (Figure 1). PH affects up to 31% of patients with clinically significant SSc-ILD and results in higher mortality than in SSc-ILD patients without PH [30]. The mechanisms for increased PVR are multifactorial. The main contributor is likely chronic hypoxic vasoconstriction, but vascular remodeling may occur, even in the absence of hypoxemia, suggesting that alternative mechanisms such as inflammation, cell proliferation involving all three components of the vascular wall as well as parenchymal loss because of fibrosis may independently contribute to PH in these patients [31]. Although the etiology of PH is associated with the extent of lung disease, the mPAP does not seem to correlate with the extent of fibrosis on imaging or the FVC [30,32]. Up to 25% of these patients have an mPAP >35 mm Hg, suggesting that their PH is “out of proportion” with what would be expected from ILD alone. This finding suggests a high probability of intrinsic vascular disease, PAH, in addition to ILD-PH alone [4,9,33]. Several studies have shown that the association of PH and ILD increases the mortality risk up to five-fold over SSc-PAH [34,35]. A recent randomized clinical trial, INCREASE, studied the effects of inhaled Treprostinil vs. Placebo in participants with PH and ILD, including SSc patients, and demonstrated a statistically significant increase, at week 16, in exercise capacity reflected by improvement in the 6-min walk distance in the treatment arm (31.12 m, 95% CI, 16.85 to 45.39; *p* < 0.001). In addition, there was an associated lower rate of clinical worsening events (HR, 0.61; 95% CI, 0.40 to 0.92; *p* = 0.04) and reduction in markers of right ventricular dysfunction [36].

### 3.3. Cardiac Involvement in SSc-PAH

SSc exerts a primary inflammatory effect on the cardiovascular system, causing myocardial fibrosis, impaired microcirculatory function, fibrosis of the conduction system leading to arrhythmias, microvascular and atherosclerotic coronary vessel disease, and hypertensive crisis [37,38,39]. Additionally, considering an older age for SSc patients as compared to IPAH, SSc often causes diastolic dysfunction with preserved ejection fraction, systolic heart failure (18% vs. 2% of SSc patients, respectively), and left atrial enlargement [39,40]. Although findings consistent with tamponade are uncommon, the presence of pericardial effusion occurs three times more often in SSc-PAH than in IPAH [6]. RV involvement is also an issue. In preclinical models, the transition from RV adaptation to RV failure is linked to diminished angiogenesis within the hypertrophic RV [41] and decreased myocardial capillary density in the failed ventricle [42]. To support these findings, and the potential role of microvascular disease, our group conducted a study to assess myocardial perfusion reserve (MPR) in SSc-PAH. MPR indices, for both RV and left ventricle (LV) were significantly lower in the PAH group compared to the scleroderma non-PAH and healthy matched control groups. Furthermore, RV and LV MPR indices were inversely associated with mPAP and RV stroke work index, as well as other measures of RV workload, systolic function, and remodeling. These data suggest that reduced myocardial perfusion may contribute to poor RV performance in patients with SSc-PAH, and potentially disease severity and worse clinical outcomes [23,43].

### 3.4. Clinical Features

Patients with SSc-PAH tend to be asymptomatic early in the disease process. They may experience progressive dyspnea, fatigue, or chest palpitations but are frequently unaware of their symptoms. Patients usually seek medical attention when they start experiencing concerning symptoms, such as severe exertional dyspnea, lightheadedness or orthostasis, and angina or chest pain [16]. Physical examination findings may be absent early in the disease, however, signs of RV dysfunction become more evident as the disease progresses. Notable findings include decreased pulse pressure, a left parasternal heave, a loud pulmonary portion of the second heart sound, a prominent jugular a-wave, and a pulmonic or tricuspid regurgitant (TR) murmur. Extracardiac findings indicative of increased right-sided pressure include hepatomegaly, hepatojugular reflex, increasing lower-extremity edema, and abdominal ascites [16,44]. Hormonal dysfunction is also common. Pro-brain natriuretic peptide (pro-BNP), a neuropeptide released in response to ventricular stretch, is frequently elevated in SSc-PAH and appears to be significantly higher than in IPAH patients despite similar hemodynamic derangements [45]. Similarly, metabolic adaptations such as hyponatremia are more common in SSc-PAH and indicate a poor prognosis [46].

## 4. Evaluation

### 4.1. Screening for SSc-PAH

Given the high morbidity and mortality related to the diagnosis of PAH in patients with SSc, several organizations and societies have designed screening algorithms that include, among others, a clinical evaluation, PFTs, laboratory tests, and echocardiography. Data suggest that SSc-PAH patients who are diagnosed and treated early as a result of screening have improved survival as compared to patients diagnosed following clinical suspicion [47,48]. The following annual strategies have demonstrated similar sensitivity, specificity, positive and negative predictive values [49]. The DETECT algorithm combines two steps. First, a “non-echocardiography” scored assessment of FVC/DLCO, telangiectasias, anticentromere antibody, pro-BNP, and EKG with right axis deviation. If the score is more than 300 points, patients receive a transthoracic echocardiography (TTE) to assess the right atrium area and tricuspid regurgitant velocity (TVR) [50]. The European Society of Cardiology/European Respiratory Society (ESC/ERS) recommends annual echocardiography that focuses on the assessment of TVR and other measurements of abnormal RV morphology and function to classify patients within a risk category [51]. Finally, the Australian Scleroderma Interest Group (ASIG) advocates for screening with pro-BNP and PFTs [52]. All these strategies have well-defined thresholds for when to refer patients for an RHC. The implementation of these recommendations still varies across institutions and clinicians. At our center, we screen SSc patients annually for PAH, and as a result, nearly 50% of new diagnoses are World Health Organization (WHO) functional class (FC) I or II [21], a dramatic difference compared with other studies that reported that 70% of cases are diagnosed as WHO FC III or IV [16,53,54] (Figure 2).

### 4.2. Echocardiography

Several studies have demonstrated a reasonable correlation between PASP estimated by TTE versus RHC, but there are some limitations to these echocardiographic techniques, including occasional poor visualization of the regurgitation envelope and operator-related technical problems [55]. TTE estimation of TR jet velocity is a widely used screening tool. This measurement correlates well with RHC-derived PASP across populations (correlation coefficient, 0.70; 95% CI, 0.67–0.73), and it is used in standardized screening tools [56,57]. TR jet velocity estimates suffer from considerable inter-operator variability, may be overestimated in patients without PH, and are often complicated by an inadequate Doppler signal [51]. Based on the ESC/ERS guidelines, PH is likely if the TR jet velocity is >3.4 m/s, as well as when values are between 2.9 and 3.4 m/s with other associated TTE findings suggestive of PH, including RV enlargement, flattening of the interventricular septum, RV outflow Doppler acceleration <105 m/s, early diastolic pulmonary regurgitation velocity >2.2 m/s, pulmonary artery diameter >2.5 cm, inferior vena cava enlargement, and right atrial area in end-systole >18 cm^2^ [51]. Although these recommendations suggest that TR jet velocity <2.8 m/s with an otherwise normal TTE makes PH unlikely, we consider that other screening modalities should be used before declaring the absence of PH in high-risk patients with SSc. Alternative features of TTE that may hint at the presence of PH include the tricuspid annular plane systolic excursion (TAPSE) that offers an easily obtained, reproducible measure of PH severity and RV function that is useful in both IPAH and SSc-PAH [58]. Our group has demonstrated that TAPSE values below 1.7 cm reflect nearly a fourfold increased risk of death compared with higher values [59].

### 4.3. Laboratory

As recommended by ASIG and other organizations, measuring pro-BNP is another alternative for screening [52]. Pro-BNP levels correlate well with RHC hemodynamics, and although a normal pro-BNP level does not exclude PAH, an elevated pro-BNP >240 pg/mL has a 90% specificity for detecting the presence of SSc-PAH [60]. Nonetheless, pro-BNP measurement should not be used as an isolated measurement because an elevated pro-BNP level is not specific to SSc-PAH and can reflect other causes of cardiac dysfunction commonly seen in patients with SSc. Hence, ASIG recommends concomitant use of PFTs for screening. Moreover, pro-BNP has been demonstrated to correlate with disease severity and survival in SSc-PAH [61].

### 4.4. Pulmonary Function Test

Standard measurements (e.g., FVC, FEV1, FEV1/FVC, TLC) are useful to assess SSc lung involvement, but DLCO helps identify the presence of PH/PAH. Although it does not correlate well with RHC-derived hemodynamics, less than one-sixth of patients with SSc-PAH have a DLCO >60% predicted [62,63]. A decrease in DLCO <60% or >20% in one year in the absence of significant lung volume abnormalities, or an FVC/DLCO percent >1.6 suggests PH [12]. York et al., reported that a DLCO/alveolar volume (Va) of <70% predicted suggested an 18-fold higher risk for developing SSc-PAH within 2.5 years compared with a DLCO/Va >70% [16,49,63].

### 4.5. Cardiac MRI

CMR is being utilized with increasing frequency in the assessment of PAH, in general, and SSc-PAH in particular. Compared with echocardiography, CMR provides information on cardiac involvement in patients with SSc, including inflammatory, microvascular, and fibrotic changes, as well as a more comprehensive assessment of the RV function and interventricular dependence [64,65] (Figure 3). Hagger et al. evaluated 40 SSc patients suspected of having PAH who underwent CMR and RHC. The ventricular mass index (VMI) was found to have a strong positive correlation with mPAP (r = 0.79, *p* < 0.01) and PVR (r = 0.8, *p* < 0.01), and a moderate negative correlation with cardiac index (CI) (r =−0.65, *p* < 0.01) [66,67]. As a diagnostic technique, the role of CMR is yet undefined, but as a prognostic tool, our group has validated CMR as a reliable instrument to assess response to therapy [68].

### 4.6. RHC and Vasodilator Testing

RHC is the gold standard for the diagnosis of PAH. SSc-PAH patients typically have less severe hemodynamic impairment compared to IPAH [6]. Weathearld et al., analyzed data from 513 patients with incident SSc-PAH enrolled in the French Pulmonary Arterial Hypertension Network Registry (FPHN) and reported important prognostic hemodynamic factors, including cardiac index, stroke volume index (SVI), pulmonary arterial compliance, and PVR [69]. We previously reported similar findings in a smaller cohort of SSc-PAH patients [21]. During RHC, it is standard to evaluate the response of the pulmonary arteries to the administration of acute vasodilators (e.g., inhaled nitric oxide or intravenous adenosine). About 10% of IPAH patients have a vasodilator response at the time of RHC, defined on strict criteria [70] (reduction of mPAP ≥10 mmHg to an absolute value ≤40 mmHg accompanied by an increase or no change in cardiac output), and such patients should be treated with high-dose calcium channel blockers (CCBs) [51]. In SSc-PAH, far fewer patients (about 1%) demonstrate vasodilator responsiveness [71,72]. A recent study compared the DETECT algorithm with the 2009 ESC/ERS guidelines and post hoc with the 2015 ESC/ERS guidelines in 195 SSc patients. Of the three patients who were diagnosed with PAH, all three algorithms had recommended RHC. The DETECT algorithm referred the most patients for RHC, but the positive predictive value was only 6% compared with 18% using the 2009 ESC/ERS guidelines alone and 23% when combining the two. The cost per patient was also highest when using DETECT alone [41,73].

## 5. Management

### 5.1. Therapy

Many medications have been studied and approved for use in PAH, including patients with SSc-PAH, but despite an improved understanding of the condition, little progress has been made in modifying outcomes with the available main therapeutic modalities: prostacyclin analogs, endothelin receptor antagonists (ERAs), and phosphodiesterase inhibitors. Apart from targeted therapy, patients benefit from supplemental oxygen (if indicated), diuretics for the management of volume overload, and treatment of atrial arrhythmias [44].

After confirmation of the diagnosis of SSc-PAH, the 6th World Symposium on Pulmonary Hypertension recommends initiating PAH targeted therapy based on risk stratification. The main objective is to achieve a low-risk status that is associated with reduced mortality (annual mortality of <5%) [74,75]. The two approaches most commonly used to assess risk are the FPHN and REVEAL 2.0. The FPHN risk assessment combines clinical and hemodynamic parameters [76] whereas the REVEAL 2.0 risk score comprises a larger number of variables providing greater risk discrimination than the FPHN [77] (Table 1). Patients with low to intermediate risk are started on combination therapy, with a few exceptions in which monotherapy is an adequate alternative. Choice of medication is usually based on a number of factors, including comorbidities, side effects, route of administration, and patient preference. High-risk patients should be treated with combination therapy that includes a parenteral prostacyclin analog [75]. Patients are usually monitored within 1 to 3 months after initiating therapy to evaluate treatment response and thereafter, every 3 to 6 months, depending on patient stability [16,49]. Tests and evaluations recommended during follow-up include clinical assessment, WHO FC, 6MW, pro-BNP, and echocardiogram. A RHC should be considered three to six months after initiation or change in therapy, and yearly thereafter [51]. Treatment should be escalated in patients who fail to achieve a low-risk status within three to six months. Those failing triple therapy should be considered for lung transplantation or palliative care [75] (Figure 2).

#### 5.1.1. Prostacyclin Analogs

Epoprostenol, treprostinil, and iloprost have been approved for the treatment of SSc-PAH [52]. Epoprostenol is given intravenously due to its short half-life. Treprostinil may be administered intravenously, subcutaneously, orally, or by inhalation. Iloprost is only administered by inhalation in the US, although an intravenous formulation is available in Europe. Epoprostenol has not demonstrated a survival advantage in this patient population; however, it improves exercise capacity, functional class, and hemodynamics (decrease in mPAP and PVR) [78]. Treprostinil offers the option of subcutaneous administration, however, patients frequently experience infusion site skin irritation, limiting its tolerability. A randomized trial of continuous subcutaneous treprostinil in 470 PAH patients (including 17% connective tissue disease CTD-PAH) demonstrated improved exercise tolerance, dyspnea indices, and hemodynamics, although only half of the CTD-PAH group had SSc [79,80]. Intermittent intravenous iloprost infusion has also demonstrated benefits in SSc patients, decreasing sPAP and improving 6MW distance [81]. Still, despite the potential efficacy of prostacyclin agents, the need for continuous infusion, meticulous catheter care, and daily preparation of the medication can be challenging in patients whose manual dexterity may be impaired by significant Raynaud’s phenomenon, sclerodactyly, and digital ulcerations. Although inhaled prostacyclin analogs have been developed to treat PAH, studies in SSc-PAH patients are limited. Encouraging results from the INCREASE trial, where SSc-ILD patients were given inhaled treprostinil for the treatment of PH, are opening the door for new trials directed to other subgroups of SSc-associated PH [36]. The utility of inhaled formulations is limited by the frequency with which the medication must be dosed (4 times a day and up to 9 inhalations each time) [44]. The oral prostacyclin receptor agonist, selexipag, does not have these limitations. In a subgroup analysis of CTD-PAH patients from the GRIPHON study, selexipag delayed disease progression of PAH, defined as a reduction in functional capacity or need for additional therapy, and was well tolerated [82,83].

#### 5.1.2. Phosphodiesterase Inhibitors

Phosphodiesterase type 5 inhibitors (PDE-5Is), such as sildenafil and tadalafil, offer a potential advantage over other therapies in that they are orally administered, well tolerated, and only require dosing one (tadalafil) or three times (sildenafil) daily. Data on the effectiveness of PDE-5Is in SSc-PAH have been difficult to interpret because SSc-PAH is not well represented in the clinical trial study cohorts [16]. SUPER-1/SUPER-2 demonstrated the efficacy of sildenafil in improving 6MW distance, hemodynamics (mPAP and PVR), and functional class in patients with PAH, including those with CTD-PAH (45% of whom had SSc) [84,85]. Similarly, PHIRST-1/PHIRST-2 demonstrated that tadalafil improves 6MW distance, quality of life, and reduces clinical worsening in a PAH population that included CTD-PAH patients. Although, treatment with tadalafil in patients with CTD-PAH was less efficacious than in patients with IPAH [86,87]. Despite the lack of a trial designed to explicitly assess the effect of this medication class in SSc-PAH patients, their cost, ease of administration, improvement in healing and prevention of development of digital ulcers [52], and tolerability make these agents ideal for first-line therapy in SSc-PAH.

#### 5.1.3. Endothelin Receptor Antagonists

Bosentan, ambrisentan, and macitentan are ERAs approved for the treatment of SSc-PAH. Bosentan, a dual oral receptor antagonist of the endothelin receptor type A (ETA) and endothelin receptor type B (ETB), was evaluated in the BREATHE-1 trial and demonstrated improvement in 6MW distance, hemodynamics, and time to clinical worsening in patients with IPAH [88]. However, a subgroup analysis of patients with SSc-PAH included in this study revealed a nonsignificant trend towards improvement in 6MW distance (3 m improvement in SSc-PAH vs. 46 m in IPAH) [89]. Conversely, ambrisentan, a more selective ERA with the advantage of preserving the vasodilatory effect of nitric oxide and prostacyclin released by endothelial cell endothelin-B receptors while suppressing vasoconstriction and cellular proliferation activated by endothelin-A receptors, demonstrated in ARIES-1/ARIES-2 a modest improvement in the 6MW distance and slower time to clinical worsening in CTD-PAH, though survival was better in the IPAH population [90,91]. Macitentan is the newest medication in this class and functions as a dual endothelin receptor antagonist. Macitentan was evaluated in the SERAPHIN trial and showed that, when added to background PAH therapy or placebo, it significantly reduced morbidity and mortality among patients with PAH. The results were consistent among subgroups of PAH including CTD-PAH (30% of the study population) [92,93].

#### 5.1.4. Guanylate Cyclase Stimulator

Riociguat exerts its effects as a stimulator of soluble guanylate cyclase (sGC), a key enzyme in the nitric oxide signaling pathway. Riociguat was evaluated in the PATENT-1/PATENT-2 trials and demonstrated significant efficacy in CTD-PAH, including SSc-PAH, improving the 6MW distance, hemodynamics (PVR and CI), and functional class. Although these improvements were less pronounced in patients with CTD-PAH than IPAH, the 2-year survival rates were similar in both PAH types [49,94,95]. The RIVER study revealed that long-term treatment with riociguat is associated with reduction in right heart size and improvement in RV function in patients with PAH (14% of patients with CTD-PAH) and CTEPH [96].

#### 5.1.5. Combination Therapy

Combining agents from two different classes is now the standard of care in patients with IPAH. Studies addressing this research question and subgroup analyses of landmark trials suggest similar benefits in patients with SSc-PAH. The AMBITION trial (small percentage of patients with CTD-PAH) showed that ambrisentan and tadalafil, when combined, improved hemodynamics, 6MW distance, and lowered the risk of clinical worsening when compared to monotherapy [97]. Our group conducted a prospective, open-label, multicenter trial of 24 patients with SSc-PAH to study the upfront combination (ambrisentan and tadalafil) in SSc-PAH treatment-naive patients, and demonstrated improvement in hemodynamics, 6MW distance, and RV structure and function [68]. A follow-up to this study, the ATPAHSS-O trial (SSc-PAH), showed improvement in both RV and LV function as assessed by CMR, as well as improvement in 6MW distance, pro-BNP, and hemodynamics [98]. More recently, SERAPHIN (no post hoc analysis for CTD- or SSc-PAH) and GRIPHON trials have demonstrated lower morbidity and mortality rates with the addition of macitentan and selexipag to background therapy, respectively [99,100].

### 5.2. Adjunct Therapies

Routine anticoagulation is not generally recommended in SSc-PAH unless clinically indicated for an associated condition, given that patients with SSc are at higher risk of ulcerative esophagitis and gastric antral vascular ectasias. To highlight the benefits and risk of this intervention, the COMPERA trial showed survival benefits with anticoagulation in IPAH and CTD-PAH patients, but a post hoc analysis of SSc-PAH patients demonstrated a non-significant statistical trend towards worse survival among those taking anticoagulants compared with patients not on anticoagulant therapy [101,102]. Furthermore, long-term use of warfarin was associated with a worse prognosis in patients with SSc-PAH in the REVEAL Registry [103]. To broaden our understanding of this matter, an ongoing trial, SPHInX, is evaluating the effect of apixaban in SSc-PAH on targeted therapy [104]. Corticosteroids have been shown, in observational studies, to be beneficial in patients with other types of CTD-PAH with improvement in hemodynamics and possible survival benefits. This response has not been replicated in SSc-PAH, which is refractory to this medication class [105]. Additionally, in this population, CCBs carry some risk of side effects, including exacerbation of esophageal reflux and esophageal dysmotility. Therefore, current guidelines do not promote vasodilator challenge during RHC or treatment of PAH with CCBs in SSc patients [50,51,52]. However, CCBs are often used, with caution, for treatment of Raynaud syndrome in these patients.

### 5.3. Investigational Therapies

Knowing the role of autoimmunity in the pathogenesis of SSc-PAH, rituximab, an anti-CD20 medication that targets B-cell populations and may lower platelet-derived growth factor-specific antibodies, has been identified as a potentially effective adjuvant therapy. In a recent multi-center, controlled trial, 57 SSc-PAH patients on stable-dose standard PAH therapy were randomized to receive rituximab or placebo. In the primary analysis, using data through week 24, the adjusted mean change in 6MW distance favored the treatment arm but did not reach statistical significance (23.6 ± 11.1 m vs. 0.5 ± 9.7 m, *p* = 0.12). When data through week 48 were also considered, the estimated change in 6MW distance was 25.5 ± 8.8 m for rituximab and 0.4 ± 7.4 m for placebo (*p* = 0.03). In addition, rituximab appeared to be safe and well tolerated [106]. Further work is needed to validate the potential benefit of this promising approach as adjuvant therapy. Ifetroban, a thromboxane A2/prostaglandin H2 receptor antagonist, and bardoxolone, an inductor of the nuclear factor erythroid 2-related factor 2 (Nrf2) and suppressor of the nuclear factor-kB (NF-kB), are currently under investigation for the treatment of CTD-PAH [49].

### 5.4. Lung Transplant

As in many progressive pulmonary diseases, lung transplantation is a therapy of last resort for many patients. Unfortunately, patients with SSc are generally considered poor candidates for lung transplantation because of an increased risk of aspiration from esophageal dysmotility, renal disease from nephrotoxic immunosuppressants, severe Raynaud phenomenon, non-healing digital ulcerations that pose a risk of infection, and very rarely, severe chest wall skin thickening leading to restriction [107]. However, various studies have shown that patients with SSc-PAH experience similar one- and two-year survival rates after lung transplant when compared to IPAH or idiopathic pulmonary fibrosis patients [108,109].

## 6. Survival and Prognosis

PAH is an independent risk factor for mortality among patients with SSc, and the severity of PAH predicts mortality in those with SSc-PAH. Although survival from SSc–PAH may have improved in the era of PAH-directed therapy, the poorer response to treatment compared with IPAH and CTD-PAH contributes to this trend. According to the large REVEAL registry data, among patients with CTD afflicted with PAH, three-year survival is worse in patients with SSc-PAH, averaging 61% and 51% compared to 80% and 76% for non SSc-CT-PAH patients, for prevalent and incident patients, respectively in each group [110] (Figure 4). A meta-analysis of 22 studies reported survival rates in SSc-PAH of 81%, 64%, and 52% over the first, second, and third years, respectively. Baseline hemodynamic measurements of PAH severity were significantly correlated with this trend. Although, there is no single outcome measure sufficiently powerful to generate an accurate assessment of prognosis or therapeutic success in this patient population. Age, male sex, DLCO, pericardial effusion, 6MW distance, and specific RHC indicators are considered significant prognostic factors [69,111] (Table 2).

## 7. Conclusions

In summary, there are several complications associated with SSc, but among them, the pulmonary vascular disease spectrum portends the most devastating prognosis. PH can present in a variety of flavors depending on the underlying mechanism, including PH related to remodeling of pre-capillary arterioles (PH group 1/SSc-PAH), PH associated with left heart disease (PH group 2), and PH due to chronic hypoxia from advanced ILD (PH group 3). Group 4 (related to chronic thromboembolic PH) is rarely encountered in this population but can certainly complicate this syndrome as well. Screening and risk stratification have evolved throughout the years, including specific laboratory, imaging, hemodynamic, and ancillary assessments, leading to a shift in the focus to early diagnosis and therapy initiation in this population at risk. Dual upfront combination therapy has certainly been a significant therapeutic advance in this disease and offers new hopes for SSc-PAH [68,97] which, however, remains the leading cause of morbidity and mortality. Whether specific targeting of immune pathogenic mechanisms may be the key for better outcomes remains unclear at this time [106]. It is also possible that a better understanding of fibrotic processes (a key pathologic feature in this syndrome) may lead to targeted therapy with drugs such as tyrosine kinase inhibitors which are currently being tested in clinical PAH trials. Hopefully, we are entering a new dawn for a more effective treatment of SSc-PAH.

## Figures and Tables

**Figure 1 diagnostics-11-00911-f001:**
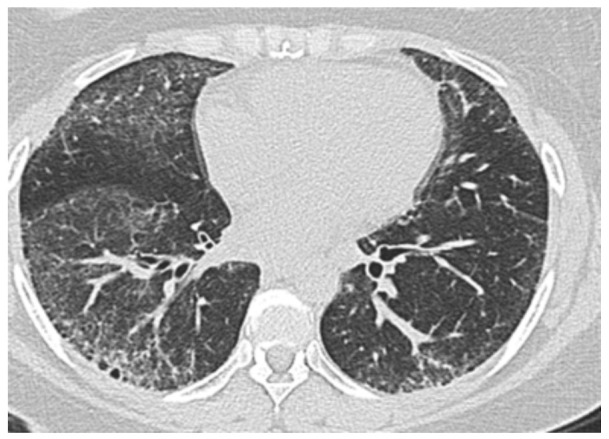
High-resolution chest CT image demonstrating bilateral peripheral interlobular septal thickening and cystic changes, more predominant in the lower lobes, compatible with systemic sclerosis interstitial lung disease (SSc-ILD) associated pulmonary hypertension (PH).

**Figure 2 diagnostics-11-00911-f002:**
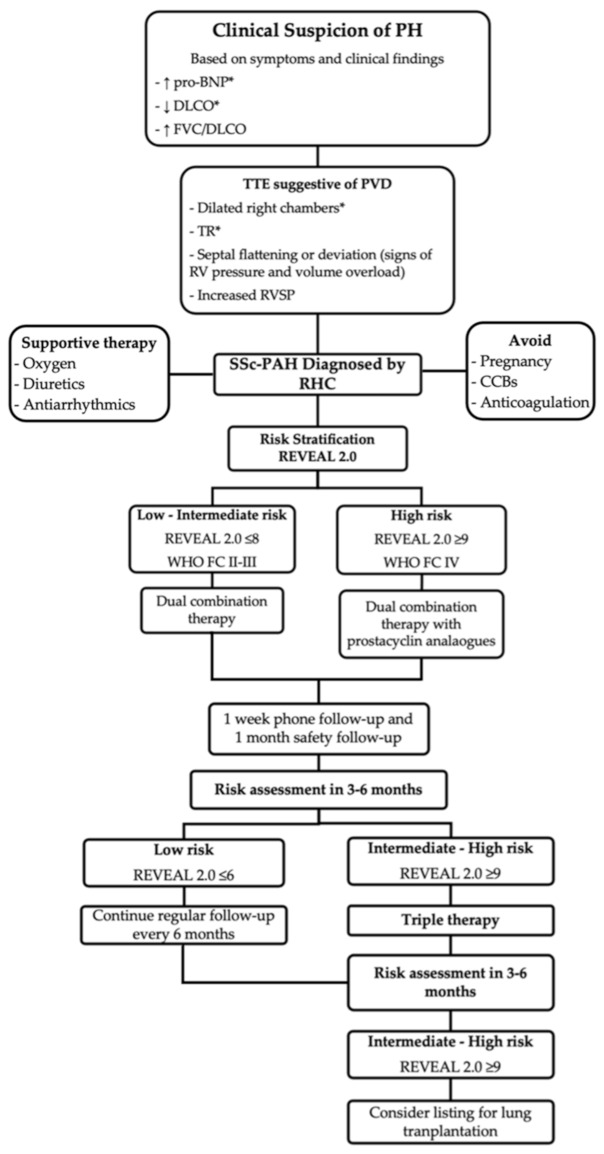
Screening, diagnostic, and treatment algorithm for systemic sclerosis-associated pulmonary arterial hypertension (SSc-PAH) used in the Johns Hopkins Pulmonary Hypertension Program. Variables based on the * DETECT algorithm, including cutoffs for referral for RHC, and REVEAL risk stratification scoring system to guide therapeutic management. Abbreviations: pro-BNP, pro brain natriuretic peptide; DLCO, diffusing capacity for carbon monoxide; FVC, forced vital capacity; TTE, transthoracic echocardiography; PVD, pulmonary vascular disease; TR; tricuspid regurgitation; RVSP; right ventricular systolic pressure; RHC, right heart catheterization; WHO FC, World Health Organization Functional Class [16,49,50].

**Figure 3 diagnostics-11-00911-f003:**
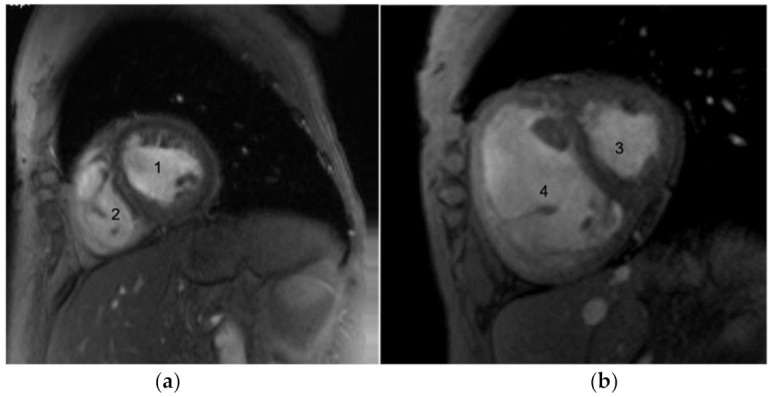
Cardiac MRI image showing: (**a**) normal left ventricle (LV) (1) and right ventricle (RV) (2) chambers in contrast with (**b**) an abnormal LV septal flattening with small LV chamber (3) and associated enlarged RV (4), which are suggestive of elevated right heart volume and pressure, which is typical in the setting of PAH.

**Figure 4 diagnostics-11-00911-f004:**
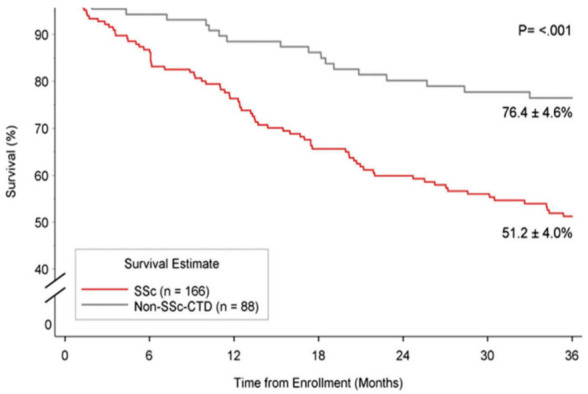
Kaplan–Meier three-year survival curves comparing patients with SSc-PAH (SSc) vs. PAH associated with connective tissue disease (CTD-PAH) (non-SSc-CTD) among participants enrolled in the REVEAL registry. Figure reproduced with permission from Elsevier, Chung et al., Chest (2014) 146(6):1494–1504 [110], Open access content.

**Table 1 diagnostics-11-00911-t001:** PAH risk scoring systems to guide management.

	REVEAL 2.0	FPHN
**Indicators**	PAH etiology (CTD-PAH, PoPH, Heritable)	WHO FC (I/II)
	Demographics (Male > 60 years)	6MWD (>440 m)
	Comorbidities (eGFR < 60 mL/min/1.73 m^2^)	RAP (<8 mmHg)
	NYHA/WHO FC (III, IV)	CI (≥2.5 L/min/m^2^)
	Vital signs (SBP < 100 mmHg, HR > 96 BMP)	
	All-cause hospitalizations ≤ 6 months	
	6MWD (<165 m)	
	Pro-BNP (>1100 pg/mL)	
	Echocardiogram (Pericardial effusion)	
	PFT (DLCO < 40% predicted)	
	RHC (mPAP > 20 mmHg within 1 year)	
**Risk score**	Low: ≤6	Low: 3–4
	Intermediate: 7–8	Intermediate: 2
	High: ≥9	High: 0–1

**Abbreviations:** CTD-PAH, PAH associated with connective tissue disease; PoPH, Porto-pulmonary hypertension associated PAH; eGFR, estimated glomerular filtration rate; NYHA, New York Heart Association; WHO FC, World Health Organization functional class; SBP, systolic blood pressure; HR, heart rate; BMP, beats per minute; 6MWD, six-minute walk distance; pro-BNP, pro brain natriuretic peptide; PFT, pulmonary function test; DLCO, diffusing capacity for carbon monoxide; RHC, right heart catheterization; RAP, right atrial pressure; CI, cardiac index [8,49,77].

**Table 2 diagnostics-11-00911-t002:** Prognostic factors for adverse outcomes in SSc-PAH.

Prognostic Factors
Male gender
Age > 60 years
WHO functional class IV
6MWD < 165 m
Anti-U1 ribonucleoprotein (RNP) negative status
DLCO < 50% predicted
Pericardial effusion
RA pressure > 20 mmHg
PVR > 32 Wood units
SBP <110 mmHg

**Abbreviations:** WHO, World Health Organization; 6MWD, six-minute walk distance; DLCO, diffusing capacity for carbon monoxide; RA, right atrial; PVR, pulmonary vascular resistance; SBP, systolic blood pressure [9,111].

## Data Availability

No new data were created or analyzed in this study. Data sharing is not applicable to this article.

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
