# Peer review of "Systemic Sclerosis-Associated Pulmonary Hypertension: Spectrum and Impact"

_diagnostics, 2021, doi:10.3390/diagnostics11050911_

Round 1
Reviewer 1 Report
This is an elegantly written, comprehensive review on SSc-PAH.
The Authors summarized the current approach to PAH in SSc patients, and described its epidemiology, evaluation and management.
To improve the value of the review I would add a diagnostic and therapeutic flow-chart or a table summary of the current available treatments.
Author Response
Reviewer 1:
Comment 1: “To improve the value of the review I would add a diagnostic and therapeutic flow-chart or a table summary of the current available treatments.”
Reply:
- We appreciate the reviewer’s feedback. We have assembled a screening, diagnostic, and therapeutic algorithm based on the workflow at our center and recent guidelines (line 201).
Figure 2. Screening, diagnostic, and treatment algorithm for SSc-PAH used in the Johns Hopkins Pulmonary Hypertension Program. Variables based on the * DETECT algorithm, including cutoffs for referral for RHC, and REVEAL risk stratification scoring system to guide therapeutic management. Abbreviations: pro-BNP, Pro Brain Natriuretic Peptide; DLCO, Diffusing Capacity for Carbon Monoxide; FVC, Forced Vital Capacity; TTE, Transthoracic Echocardiography; PVD, Pulmonary Vascular Disease; TR; Tricuspid Regurgitation; RVSP; Right Ventricular Systolic Pressure; RHC, Right Heart Catheterization; WHO FC, World Health Organization Functional Class [16, 49, 50, 77].
Reviewer 2 Report
Authors nicely summarized the current knowlegde about systemic sclerosis-associated pulmonary arterial hypertension, an indeed catastrophic complication of this well known autoimmune diseases.
Introduction: Although not the focus of your work please elaborate in bit more detail the other forms of PH that may exist in SSc as this is the only time you mention it.
Epidemiology: The first paragraph of this section in my opinion can be shortened to some extent.
Evaluation: Part "4.6. RHC and Vasodilator Testing": Lines 277-280: Authors state "Additionally, in this population, CCBs carry some risk of side effects, including exacerbation of esophageal reflux and esophageal dysmotility. Therefore, current guidelines do not promote vasodilator challenge during RHC or treatment of PAH with CCBs in SSc patients [50-52]. However, CCBs are often used for treatment of Raynaud syndrome in these patients". --While the content of this topic is absolutely correct, it should not be placed in the Evaluation but more in the Discussion part of this Review.
Therapy Section: Great overview of the current possibilities to treat SSC-PAH, nothing to add.
Survival and Prognosis Section: I encourage the authors to discuss the available literature in more detail. Specially the work from Lefevre et al (Ref. 111) deserves more attention and discussion.
Conclusion: I am to some extent disappointed by this Section. After providing such a fine overview of the SSc-PAH topic the current mini-conclusion downgrades the overall impression of this manuscript. Please put some more work in it.
Author Response
Reviewer 2:
Comment 1: “Introduction: Although not the focus of your work please elaborate in bit more detail the other forms of PH that may exist in SSc as this is the only time you mention it.”
Reply:
- We appreciate the reviewer’s comment; however, we do address fairly extensively other pulmonary hypertension types related to SSc in the Pathogenesis section and in more detail in the Interstitial Lung Disease-associated PH in SSc, and Cardiac Involvement in SSc-PAH sections.
Comment 2: “Epidemiology: The first paragraph of this section in my opinion can be shortened to some extent.”
Reply:
- Presenting the new definition of PH/PAH is essential and relevant for the reader because the background knowledge of the audience can be variable. We would, therefore, prefer to keep this section as is.
Comment 3: “Evaluation: Part "4.6. RHC and Vasodilator Testing": Lines 277-280: Authors state "Additionally, in this population, CCBs carry some risk of side effects, including exacerbation of esophageal reflux and esophageal dysmotility. Therefore, current guidelines do not promote vasodilator challenge during RHC or treatment of PAH with CCBs in SSc patients [50-52]. However, CCBs are often used for treatment of Raynaud syndrome in these patients". --While the content of this topic is absolutely correct, it should not be placed in the Evaluation but more in the Discussion part of this Review.”
Reply:
- We agree with the reviewer’s recommendation, and we have moved lines 277 to 280 to the Adjunct therapies section (lines 465 to 469), as follow:
- Adjunct therapies: Routine anticoagulation is not generally recommended in SSc-PAH unless clinically indicated for an associated condition, given that patients with SSc are at higher risk of ulcerative esophagitis and gastric antral vascular ectasias. To highlight the benefits and risk of this intervention, the COMPERA trial showed survival benefits with anticoagulation in IPAH and CTD-PAH patients, but a post-hoc analysis of SSc-PAH patients demonstrated a non-significant statistical trend towards worse survival among those taking anticoagulants compared with patients not on anticoagulant therapy [101, 102]. Furthermore, long-term use of warfarin was associated with a worse prognosis in patients with SSc-PAH in the REVEAL Registry [103]. To broaden our understanding of this matter, an ongoing trial, SPHInX, is evaluating the effect of apixaban in SSc-PAH on targeted therapy [104]. Corticosteroids have been shown, in observational studies, to be beneficial in patients with other types of CTD-PAH with improvement in hemodynamics and possible survival benefits. This response has not been replicated in SSc-PAH, which is refractory to this medication class [105]. Additionally, in this population, CCBs carry some risk of side effects, including exacerbation of esophageal reflux and esophageal dysmotility. Therefore, current guidelines do not promote vasodilator challenge during RHC or treatment of PAH with CCBs in SSc patients [50-52]. However, CCBs are often used, with caution, for treatment of Raynaud syndrome in these patients.
Comment 4: “Therapy Section: Great overview of the current possibilities to treat SSC-PAH, nothing to add.”
Reply:
- We appreciate the reviewer’s comment
Comment 5: “Survival and Prognosis Section: I encourage the authors to discuss the available literature in more detail. Specially the work from Lefevre et al (Ref. 111) deserves more attention and discussion.”
Reply:
- We have modified this section based on the reviewer’s comment (lines 506 to 510), as follow:
- Survival and Prognosis: PAH is an independent risk factor for mortality among patients with SSc, and the severity of PAH predicts mortality in those with SSc-PAH. Although survival from SSc–PAH may have improved in the era of PAH-directed therapy, the poorer response to treatment compared with IPAH and CTD-PAH contributes to this trend. According to the large REVEAL registry data, among patients with CTD afflicted with PAH, three-year survival is worse in patients with SSc-PAH, averaging 61% and 51% compared to 80% and 76% for non SSc-CTD-PAH patients, for prevalent and incident patients, respectively in each group [110] (Figure 3). A meta-analysis of 22 studies reported survival rates in SSc-PAH of 81%, 64%, and 52% over the first, second, and third years, respectively. Baseline hemodynamic measurements of PAH severity were significantly correlated with this trend. Although, there is no single outcome measure sufficiently powerful to generate an accurate assessment of prognosis or therapeutic success in this patient population. Age, male sex, DLCO, pericardial effusion, 6MW distance, and specific RHC indicators are considered significant prognostic factors [69, 111] (Table 2).
Comment 6: “Conclusion: I am to some extent disappointed by this Section. After providing such a fine overview of the SSc-PAH topic the current mini-conclusion downgrades the overall impression of this manuscript. Please put some more work in it.”
Reply:
- We have completely modified this section based on the reviewer’s comment (lines 537 to 555), as follow:
- Conclusion: In summary, there are several complications associated with SSc, but among them, the pulmonary vascular disease spectrum portends the most devastating prognosis. PH can present in a variety of flavors depending on the underlying mechanism, including PH related to remodeling of pre-capillary arterioles (PH group 1/SSc-PAH), PH associated with left heart disease (PH group 2), and PH due to chronic hypoxia from advanced ILD (PH group 3). Group 4 (related to chronic thromboembolic PH) is rarely encountered in this population but can certainly complicate this syndrome as well. Screening and risk stratification have evolved throughout the years, including specific laboratory, imaging, hemodynamic, and ancillary assessments, leading to a shift in the focus to early diagnosis and therapy initiation in this population at risk. Dual upfront combination therapy has certainly been a significant therapeutic advance in this disease and offers new hopes for SSc-PAH [68, 97] which, however, remains the leading cause of morbidity and mortality. Whether specific targeting of immune pathogenic mechanisms may be the key for better outcomes remains unclear at this time [106]. It is also possible that a better understanding of fibrotic processes (a key pathologic feature in this syndrome) may lead to targeted therapy with drugs such as tyrosine kinase inhibitors which are currently being tested in clinical PAH trials. Hopefully we are entering a new dawn for a more effective treatment of SSc-PAH.